# CausalRivers - Scaling up benchmarking of causal discovery for real-world time-series

**Gideon Stein, Maha Shadaydeh, Jan Blunk, Niklas Penzel, Joachim Denzler**
Computer Vision Group Jena
Friedrich Schiller University Jena
Jena, Thuringia 07743, Germany
`gideon.stein@uni-jena.de`

## Abstract

Causal discovery, or identifying causal relationships from observational data, is a notoriously challenging task, with numerous methods proposed to tackle it. Despite this, in-the-wild evaluation of these methods is still lacking, as works frequently rely on synthetic data evaluation and sparse real-world examples under critical theoretical assumptions. Real-world causal structures, however, are often complex, evolving over time, non-linear, and influenced by unobserved factors, making it hard to decide on a proper causal discovery strategy. To bridge this gap, we introduce **CausalRivers**[1], the largest in-the-wild causal discovery benchmarking kit for time-series data to date. CausalRivers features an extensive dataset on river discharge that covers the eastern German territory (666 measurement stations) and the state of Bavaria (494 measurement stations). It spans the years 2019 to 2023 with a 15-minute temporal resolution. Further, we provide additional data from a flood around the Elbe River, as an event with a pronounced distributional shift. Leveraging multiple sources of information and time-series meta-data, we constructed two distinct causal ground truth graphs (Bavaria and eastern Germany). These graphs can be sampled to generate thousands of subgraphs to benchmark causal discovery across diverse and challenging settings. To demonstrate the utility of CausalRivers, we evaluate several causal discovery approaches through a set of experiments to identify areas for improvement. CausalRivers has the potential to facilitate robust evaluations and comparisons of causal discovery methods. Besides this primary purpose, we also expect that this dataset will be relevant for connected areas of research, such as time-series forecasting and anomaly detection. Based on this, we hope to push benchmark-driven method development that fosters advanced techniques for causal discovery, as is the case for many other areas of machine learning.

## 1 Introduction

Causal discovery, the process of identifying causal relationships from observational data, has made significant theoretical progress over the past decade (Pearl, 2009), (Peters et al., 2017). This has led to the development of various methods (Vowels et al., 2022), (Assaad et al., 2022) that especially bear potential for fields where randomized controlled trials are impractical due to restrictions concerning interventions, such as earth sciences, neuroscience, and economics. However, despite this progress, causal discovery remains a predominantly theoretically motivated area of research. We argue that one of the primary reasons for this is the challenge practitioners face in selecting appropriate causal discovery strategies, especially given the strong assumptions these methods are often required to make about the underlying data, e.g. causal sufficiency, linearity, or the absence of hidden confounders. As an example, methods based on additive noise models (ANMs, (Peters et al., 2011)) assume specific noise distributions, while constraint-based approaches like PC (Spirtes et al., 2001) and FCI (Spirtes, 2001) assume that causal relationships underlying observational data are of a faithful nature, an assumption that was criticized by Andersen (2013).

---

[1]https://causalrivers.github.io

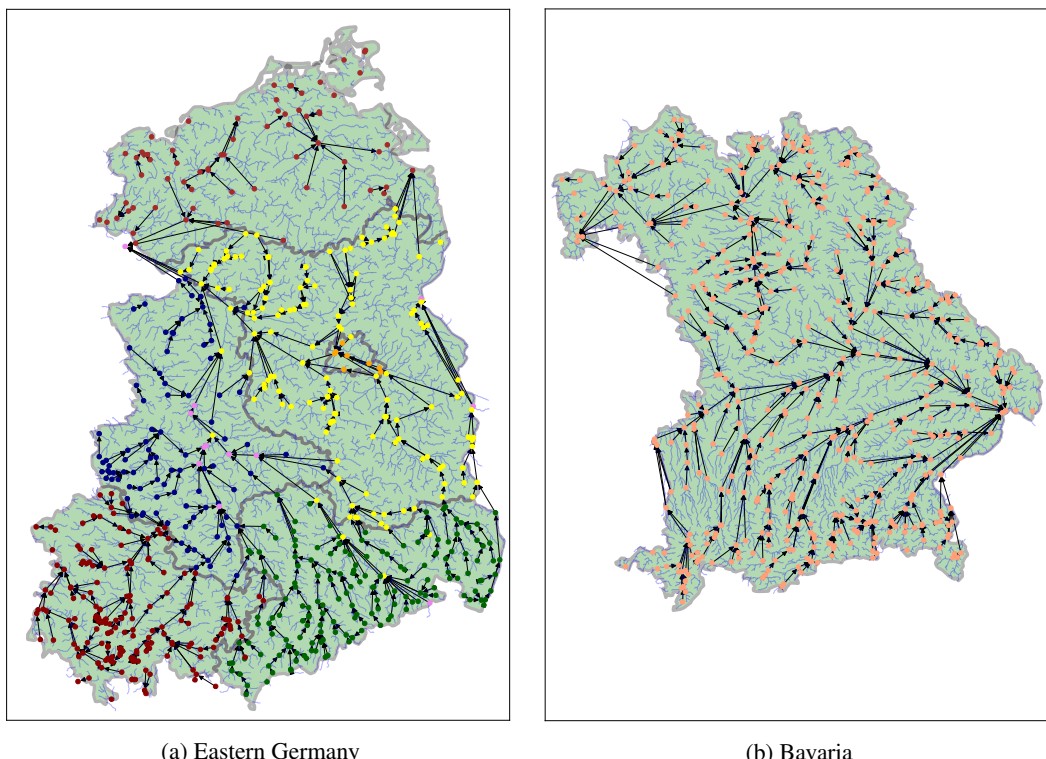

(a) Eastern Germany                (b) Bavaria

Figure 1: The causal ground truth graphs for river discharge measurement stations are provided with this benchmarking kit. Jointly, these two graphs hold over 1000 nodes. Different colors represent different data origins that we specify in appendix A.1.

Violations of these assumptions are particularly very common in fields like neuroscience or climate science, where the data-generating process is complex, often unknown, and typically influenced by unobserved confounding factors. This, in turn, also limits the reliability of synthetic benchmarking, as data-generating processes fail to meet the complexity of real-world scenarios, leading to inflated assessments of method performance, as discussed in Reisach et al. (2021). Additionally, even extensive survey papers like Vowels et al. (2022) can provide limited guidance for practitioners, as they cannot directly address which methods might provide meaningful insights when assumptions are violated. Furthermore, a large part of the causal discovery literature relies on either purely synthetic experiments (Pamfil et al., 2020) and simple real-world examples with few nodes (Mooij et al., 2016; Runge et al., 2019). This situation seems to be especially pronounced for time-series data, as even fewer datasets are available. Instead, the focus of many works lies on proving theoretical guarantees under assumptions as proof of their validity. While these insights are by no means unnecessary and provide an essential foundation for methods evaluation, they provide, again, limited help when faced with the complexity and unpredictability of the real world. Here we feel it necessary to recall the iron rule of explanation as the cornerstone of modern science (Strevens, 2020): *"scientists [...] resolve their differences of opinion by conducting empirical tests"*. In machine learning, this is implemented through benchmark datasets, which provide standardized environments for rigorous evaluation of the performance of competing methods. These benchmarks not only facilitate fair comparisons but also reveal systematic weaknesses and, thus, actively contribute to method development. For instance, computer vision was reshaped by the ImageNet challenge that brought the surprising performance of the AlexNet architecture to the field's attention (Alom et al., 2018). In a similar vein, we believe that a large-scale and realistic benchmark dataset for causal discovery could have a profound impact on the field. We also find that no such benchmark has been established for causal discovery from time-series for which we provide evidence in the next section.

To bridge this gap, and inspired by a single five-node example in Muñoz-Marí et al. (2020), we introduce **CausalRivers**, the by far largest in-the-wild causal discovery benchmarking kit, specifically

for time-series data, to date. CausalRivers features an extensive dataset on river discharge, spanning from the year 2019 to the end of 2023, with a 15-minute resolution. It covers the entirety of the eastern German territory (666 measurement stations) and the state of Bavaria (494 measurement stations). Further, we include an additional dataset from a subset of stations (A.3), which exhibits a pronounced distributional shift through a very recent extreme precipitation event (Figure 2). To complement this dataset, we constructed two causal ground truth graphs (Figure 1), that include all measurement stations. For this, we leveraged multiple informational sources such as Wikipedia crawls and remote sensing. Further information on the data origins is included in appendix A.1. Importantly, as the full ground truth graphs hold over 1000 nodes, a direct application of causal discovery approaches to these time-series is unfeasible. Instead, we provide sampling strategies to generate thousands of subgraphs with a flexible amount of nodes and unique graph characteristics such as single-sink nodes, root causes, hidden-confounding, or simply connected graphs. Along with the general characteristics of river discharge, which we discuss later, the dataset allows us to assess the impact of conditions such as e.g., high-dimensionality, non-linearity, non-stationarity, seasonal patterns, the presence of hidden confounding (through weather), misalignment of causal lag and sampling rate, and generally distributional shifts on method performance.

To demonstrate our benchmarking kit, we conducted three sets of experiments, providing an overview of potential benchmarking use cases. First, we provide experiments on multiple sets of subgraphs. For this, we report performances of well-known causal discovery approaches, provide naive yet effective baselines, and evaluate some recent deep learning approaches. Here, we find that simple strategies can be robust, where many causal discovery methods struggle. Second, we evaluate how the selection of specifically informative subsections of observational data can affect the performance of different methods, something that could prove helpful in real-world applications. Here we find mixed results, as a proper selection depends on the specific causal discovery approach. Finally, we provide some examples of how domain adaption might be an interesting tool to cope with the complex nature of the provided data distribution. To make usage as accessible as possible, we provide a ready-to-use benchmark package with many features[2]. With this benchmarking kit, we hope to pave the way for more benchmark-focused method development and provide the groundwork for closing the gap between causal discovery research and its potential applications. Finally, we are looking forward to seeing whether the provided data, as the amount of time-series data is extensive, might also be interesting to related disciplines such as time-series forecasting, anomaly detection, or regime and change point identification (Aminikhanghahi & Cook, 2017; Ahmad et al., 2024b). To summarize, this work provides the following contributions:

- The largest real-world benchmark for causal discovery from time-series to date.
- An evaluation of established causal discovery methods on high amounts of in-the-wild data.
- An introduction and a ready-to-use implementation of the complete benchmarking kit.

## 2 BACKGROUND

The impact of benchmarking becomes evident in various fields where large-scale and realistic datasets have driven significant advances. As already mentioned, computer vision was reshaped by the ImageNet challenge that brought the surprising performance of the AlexNet architecture to the field's attention (Alom et al., 2018). Other examples are GLUE (Wang et al., 2018), which has become a standard for evaluating natural language processing models. Next to this, the SQuAD benchmark (Pranav et al., 2016) has pushed the state-of-the-art in question-answering. Further, WMT-2014 (Bojar et al., 2014) helped with establishing Transformers (Vaswani et al., 2017) as the dominant architecture in natural language processing. Similarly, the LAION-5B dataset (Schuhmann et al., 2022) has driven the development of vision foundation models. Moreover, RESISC45 (Cheng et al., 2017) helped cement deep learning for remote-sensing scene classification. Finally, the Cityscapes benchmark (Cordts et al., 2016) has accelerated research in autonomous driving, while the CASP13 benchmark has revolutionized protein folding via AlphaFold (AlQuraishi, 2019).

In a similar vein, we believe that a large-scale and realistic benchmark dataset for causal discovery could have a profound impact on the field. To date, however, such a benchmark is lacking. To visualize this absence, we provide an overview of existing datasets that either cover real-world data

---

[2]https://github.com/causalrivers

Table 1: An extensive list of works that are used to evaluate causal discovery approaches. A ✓ for "Time" denotes that the data source is a time-series. A ✓ for "Real world" denotes that both observational data and ground truth causal graphs are not synthetic. Further, ⊘ denotes no theoretical limit on the number of variables as datasets have synthetic components. We emphasize that there is no comparable-sized benchmark for time-series data to date.

| Topic | Origin | Time | Real world | Number of variables |
|---|---|---|---|---|
| Semi-synthetic generation$^{ATE}$ | Neal et al. (2023) | ✗ | ✗ | ⊘ |
| Semi-synthetic generation$^{ATE}$ | Shimoni et al. (2018) | ✗ | ✗ | ⊘ |
| Gen expressions | Dibaeinia & Sinha (2020) | ✗ | ✗ | ⊘ |
| Production line | Göbler et al. (2024) | ✗ | ✗ | ⊘ |
| Gen expressions | Van den Bulcke et al. (2006) | ✗ | ✗ | ⊘ |
| Gen networks | Pratapa et al. (2020) | ✗ | ✗ | ⊘ |
| Visual understanding | McDuff et al. (2022) | ✗ | ✗ | ⊘ |
| Mixed challenge$^{ATE}$ | Dorie et al. (2019) | ✗ | ✗ | ⊘ |
| Mixed challenge$^{ATE}$ | Hahn et al. (2019) | ✗ | ✗ | ⊘ |
| Benchmark kit (LLM) | Zhou et al. (2024b) | ✗ | ✓ | 109 |
| Single-cell perturbation | Chevalley et al. (2022) | ✗ | ✓ | 622 |
| Mixed challenge | Guyon et al. (2008) | ✗ | ✓ | 132 |
| Cause-effect pairs | Mooij et al. (2016) | ✗ | ✓ | 100×2 |
| Congenital heart disease | Spiegelhalter et al. (1993) | ✗ | ✓ | 20 |
| Lung cancer | Lauritzen & Spiegelhalter (1988) | ✗ | ✓ | 8 |
| Food manufacturing | Menegozzo et al. (2022) | ✗ | ✓ | 34 |
| Protein signaling | Sachs et al. (2005) | ✗ | ✓ | 11 |
| Bridges | Yoram Reich (1989) | ✗ | ✓ | 12 |
| Abalons | Warwick Nash (1994) | ✗ | ✓ | 8 |
| Arrow of time | Bauer et al. (2016) | ✗ | ✓ | ⊘ |
| Pain diagnosis | Tu et al. (2019) | ✗ | ✓ | 14 |
| Aerosols | Jesson et al. (2021) | ✓ | ✓ | 14 |
| Industrial systems | Mogensen et al. (2024) | ✓ | ✓ | 233 |
| Semi-synthetic generation | Cheng et al. (2023) | ✓ | ✗ | ⊘ |
| ODE | Kuramoto (1975) | ✓ | ✗ | ⊘ |
| Gen networks | Greenfield et al. (2010) | ✓ | ✗ | ⊘ |
| FMRI | Smith et al. (2011) | ✓ | ✗ | 50 |
| Benchmark kit (CauseMe) | Muñoz-Marí et al. (2020) | ✓ | ✓/✗ | 5 / ⊘ |
| Benchmark kit (OCBD) | Zhou et al. (2024a) | ✓ | ✓/✗ | 11 / ⊘ |
| Multi-Benchmark | **CausalRivers** | ✓ | ✓ | >1000 |

or attempt to imitate specific characteristics of real-world domains (semi-synthetic data) in Table 1. For completeness's sake, we also include datasets that only provide sample data (no temporal dimension) as well as some datasets that are considered for average treatment effect estimation since it is possible to repurpose them for causal discovery. As can be observed from our summary, while we found almost 30 distinct datasets, few of them provide time-series data. Further, many datasets that provide authentic, real-world data have a limited number of nodes included, making it hard to rely on them for benchmarking as they become susceptible to overfitting. Of course, we are not the first to recognize the difficulty of benchmarking and comparisons in the causal discovery literature. Often, this situation is attributed to the fact that causal ground truth, along with proper observational data, is notoriously hard to find (Mogensen et al., 2024; Niu et al., 2024). Noteworthy, some works that attempt to improve on this situation through other means are Montagna et al. (2023), which tries to assess the robustness of causal discovery methods towards violations of their assumptions, or Faller et al. (2024), which attempts to score methods based on their consistency on multiple subsets of data. Further, some approaches such as Muñoz-Marí et al. (2020), Niu et al. (2024) or Zhou et al. (2024b), aim to provide benchmarking through a collection of varying synthetic and semi-synthetic data sources. While these approaches are, of course, a step in the right direction and should be considered along real-world benchmarking, they are not sufficient to fully dissect performance dif-

ferences of varying causal discovery methods for in-the-wild applications. Finally, as on recent and promising attempt to benchmark causal discovery performance, we want to mention Mogensen et al. (2024) as complementing work. Here, the ground truth graph is of sufficient size (Table 1) to properly benchmark performance. Importantly, as the domain is completely distinct from ours, we see this work as a promising additional benchmarking approach.

## 3 BENCHMARK DESCRIPTION

Table 2: Overview of the three provided datasets in CausalRivers.

| Name | Nodes | Edges | Start date | End date | Resolution |
|---|---|---|---|---|---|
| RiversEastGermany | 666 | 651 | 1.1.2019 | 31.12.2023 | 15min |
| RiversBavaria | 494 | 490 | 1.1.2019 | 31.12.2023 | 15min |
| RiversElbeFlood | 42 | 42 | 09.09.2024 | 10.10.2024 | 15min |

Here, we provide information on the origin of the data included in our benchmark kit and on the construction of causal ground truths. Further, we discuss unique challenges for causal discovery on in-the-wild datasets and some specific features that are native to our data domain: Hydrology. Finally, to provide a comprehensive overview, we also include a list of features that we provide next to the data in our benchmarking kit, such as sampling strategies and naive baselines.

### 3.1 BENCHMARK CONSTRUCTION

This benchmarking kit is concerned with river discharge, so the amount of water that flows through a river. It is measured in $m^3/s$. As the amount of water measured at an upstream station directly influences the amount of water measured by all downstream stations at a later point in time, we consider them as causal. Through causal discovery, these causal relationships are potentially recoverable from observational data, in this case, time-series data, alone. To produce the datasets provided in our benchmarking kit, we began by collecting information on available measurement stations in our selected geographical area. Through cooperation with eight different German state agencies (A.1, each state has its own network of measurement stations that serve primarily for flood prevention), we were provided with raw time-series data along with some measurement station metadata. After some initial filtering (mostly removing duplicates and malfunctioning measurement stations), we ended up with around 666 and 494 valid time-series for the selected time intervals. Notably, no further preprocessing was performed, as we consider it interesting to challenge researchers to come up with preprocessing steps that specifically benefit their causal discovery approach. To construct the causal ground truth for these measurement stations, we leveraged a mixture of meta-information provided by the state agencies, remote-sensing (Wickel et al., 2007), Wikipedia information crawls and handcrafting for a semi-automatic construction of the graph. Further, all edges were double-checked by hand in the final stage to correct for potential matching errors. For documentary purposes, we provide the full construction pipeline [3]. Note that it was specifically constructed to allow additional nodes to be added in the future. With this, and especially as there was recently a call for less static benchmarks (Shirali et al., 2022), we leave room to extend the provided data in the future. In summary, we provide three distinct sets of time-series as displayed in Table 2, along with two ground truth causal graphs (Figure 1), as the RiversElbeFlood causal ground truth is a subset of the RiversEastGermany graph. Importantly, we envision RiversEastGermany as the primary benchmarking source as it is more diverse in terms of geography and data origin than RiversBavaria. Alternatively, we suggest RiversBavaria as a tool for the exploration of domain adaptation.

### 3.2 BENCHMARKING KIT FEATURES

To maximize the usability of this benchmarking kit, we provide additional tools and resources along with the time-series and causal ground truth graphs. These tools and resources should allow researchers to tailor the dataset to their specific needs and evaluate the performance of methods in a more targeted and streamlined manner. Specifically, we provide:

---

[3]https://github.com/causalrivers/documentation

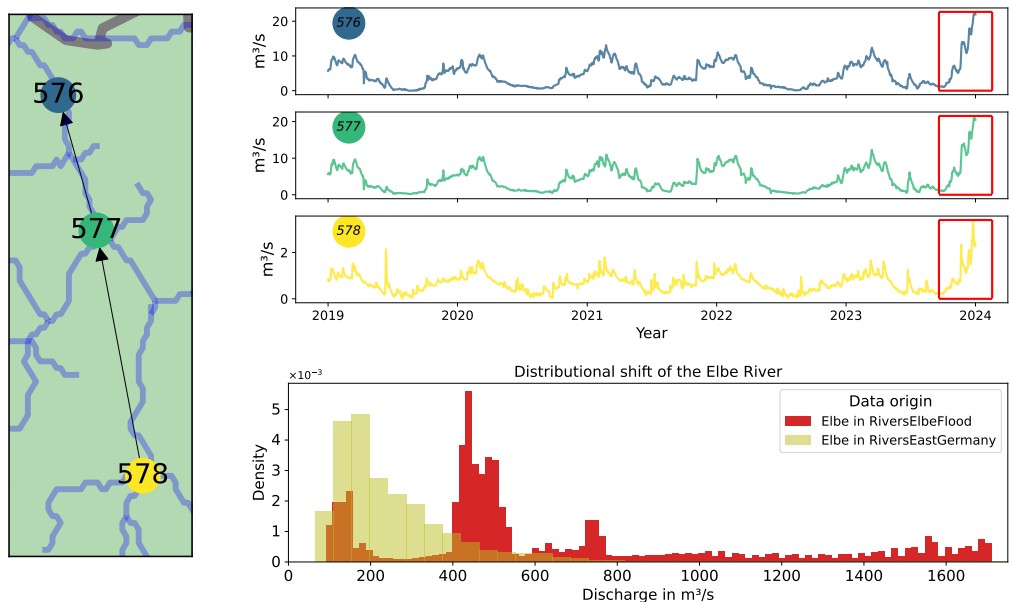

Figure 2: Left/Top: A single sampled causal relationship along with time-series data from RiversEastGermany. A massive precipitation event is marked in *red*. Right bottom: The pronounced distributional shift between the same nodes of the Elbe in RiversEastGermany and RiversElbeFlood.

- Tools to sample from ground truth causal graphs to access subgraphs with any number of nodes. Further, subgraphs can be restricted through graph characteristics such as connectivity or, e.g., geographical reality. An example of such a sample can be found in Figure 2.

- Strategies to assess climatic conditions, especially precipitation, around any node by building on the German weather service (DWD). These tools might be helpful for dissecting confounding effects and selecting specifically interesting time-series windows.

- General preprocessing, data loaders, and display tools for all included datasets.

- Three naive baseline strategies that can be used to assess performance properly.

- Tutorials on all provided tools and on how to reproduce the results reported in section 4.

## 3.3  BASELINE STRATEGIES

With our benchmarking kit, we provide three baseline causal discovery strategies. First, we determine the causal direction between two time-series, here denoted as $x_1$ and $x_2$, purely based on cross-correlation between $x_1$ and lagged versions of $x_2$. For this, we look for the lag at which the cross-correlation is maximized. If this lag is negative, meaning the highest correlation is between the present of $x_1$ and the future of $x_2$, $x_1 \longrightarrow x_2$ is inferred. If the lag is positive, $x_1 \longleftarrow x_2$ is inferred. We call this strategy simply **CC** for Cross-Correlation. Second, we rely on the actual magnitude of the time-series, featuring a principle of causality that can be found in physics, where the mass of an object determines the causal direction (e.g., gravity). While in Physics, the arrow of causation typically points at the object with the lower mass, for rivers, this is reversed, as it is technically impossible for a very big river flows in a smaller river (at least without river splits). To leverage this principle, we simply assume $x_1 \longrightarrow x_2$ if the mean of $x_2$ is bigger than the mean of $x_1$. We call this strategy Reverse Physical, in short, **RP**. Notably, both RP and CC decide on one direction for each potential edge. However, as it is typically the case that rivers only flow in a single location, we additionally restrict these strategies to select a maximum of one child node per parent node. This is done either by selecting the next larger river (+N) or the biggest river (+B) as the only link or the river with the highest cross-correlation as the successor (+C). Finally, we evaluate the union between RP and CC, which we denote **Combo** and where we also test each restriction.

### 3.4 UNIQUE CHARACTERISTICS

Because our benchmark dataset covers a large area of Germany and is combined from multiple data sources, it exhibits a number of interesting and unique features. Further, the domain of Hydrology brings in, of course, additional unique characteristics. In the following, we will discuss these attributes to help understand the complexity of the dataset. With this, we also hope to shine a light on the specific challenges and opportunities it provides for causal discovery. Further, we include some additional dataset statistics in A.1.

• **Geographical Realities:** With over 1000 nodes, the datasets cover a wide range of geographical conditions, such as mountainous, coastal, and urban areas, and a wide variety of distances between stations. With this, it also covers a wide range of causal structures, lags, and strengths. Additionally, while the geographic closeness of nodes influences the difficulty of detecting a causal relationship, other factors such as effect strength and elevation (and with this flow speed) also play a major role. The datasets include a range of interesting geographical anomalies, such as dams, pump water storages, artificial canals, and tide effects, which can affect the causal relationships between nodes by altering the flow rate, water level, and consistency of relationships.

• **Weather Confounding:** Weather confounding plays an important role in the innovation of all time-series in the dataset. Rainfall can occur in a single node, across all nodes, or in a subset of nodes. Therefore, the impact of weather might be beneficial to determine causal direction or be detrimental. Further, as rainfall appears suddenly, the dataset is characterized by non-stationarity, non-linearity, and seasonal patterns. To visualize, Figure 2 displays the effect of a massive precipitation event at the end of the time-series that affects all nodes in the sample.

• **Causal Lag:** Due to the varying distance and elevation between nodes, the speed of the rivers, and, in turn, the lag at which the causal effect occurs varies greatly throughout the dataset. Moreover, the causal lag of a specific relationship differs throughout the years as it depends on the amount of water that is present at a given time (the more water, the higher the velocity of the river.) We estimate this, along with weather confounding, to be a core challenge of the benchmark, as many causal discovery methods assume a static causal structure with a fixed lag.

• **Sampling Rate:** The sampling rate at which data is collected directly impacts the accuracy of inferred causal relationships (Gong et al., 2017; 2015). If the sampling rate is too low, critical causal interactions between variables are missed. Moreover, high-frequency sampling may increase the computational burden and result in models that overfit transient fluctuations rather than true causal interactions. As the dataset is provided in a 15-minute resolution, it allows to explore the impact of different sampling and aggregation rates on causal discovery performance in real-world applications.

• **Domain Biases:** Besides occasionally allowing for the provision of a skeleton graph (e.g. (Runge et al., 2019)), causal discovery methods typically integrate little domain knowledge. Here, we want to note that depending on the domain, this might be unnecessarily agnostic as such information, if leveraged, could be beneficial. For CausalRivers, an example of such information might be that rivers typically have a single endpoint, which makes nodes with multiple children highly unlikely. Further, the magnitude of the time-series can be beneficial as the amount of water is unlikely to reduce along the causal direction. While these biases are quite specific to Hydrology, we expect that other biases in a similar manner exist in other domains and could also be utilized there.

## 4 EXPERIMENTAL RESULTS AND DISCUSSION

To demonstrate our benchmark kit, we conducted three experiments to demonstrate use cases and to gain interesting insights into the performance of various causal discovery strategies. During these experiments, we deployed the following well-established methods from the literature: **PCMCI** with a linear conditional independence test (Runge et al., 2019), **Varlingam** (Hyvärinen et al., 2010), **Dynotears** (Pamfil et al., 2020) and a simple linear Granger causal approach (**VAR**), aiming at covering all common archetypes (Assaad et al., 2022). Further, we evaluated two recent deep-learning techniques. First, **CDMI** (Ahmad et al., 2024a), a nonlinear Granger causal approach that analyzes residuals of deep networks under knockoff interventions. Second, Causal Pretraining (**CP**, (Stein et al., 2024)), which learns a direct mapping from multivariate time-series to a causal graph from synthetic data. Notably, we specifically chose to include CP as it directly allows for domain adaption via finetuning. Additionally, we evaluated the performance of our proposed naive baselines

during all experiments. As causal discovery methods typically come with at least some Hyperparameters, we performed a rudimentary Hyperparameter search per method which we document in appendix A.2. We, however, also note that methods that require fewer Hyperparameter configurations are more likely to be successful in practice, which should be considered when comparing methods. Therefore, while we evaluate a few method-specific parameters, we often selected default parameters. For all experiments, we test different resolutions (15min - 24H). To omit the complication of finding an individual proper decision threshold, for all experiments, we chose to report the mean (over all samples) of the AUROC scores of the best-performing Hyperparameter combination as the final performance measure for a method. During this, we ignored autoregressive links as these are always present and could potentially skew results. Finally, we want to emphasize that our benchmarking strategy serves as an example of what benchmarking with CausalRivers can look like. However, benchmarking procedures should be adjusted to the aspect of method performance in focus. For example, our approach potentially overestimates performance when either a high variance of performance between different Hyperparameter combinations exists (as they cannot be properly selected without labels) or it is hard to determine decision thresholds. These are both complications that should be taken into account for real-world applications and could also be integrated into benchmarking procedures in the future.

## 4.1 EXPERIMENT SET 1 - VARYING GRAPH STRUCTURES

As the first and most extensive experiment set, we performed causal discovery on subgraphs with varying graph characteristics and with the full-time-series available. We took RiversEastGermany as the base graph for this experiment. For each set except the last one, we report results for graphs with three or five nodes. The following graph sets were evaluated:

• **Random:** We sampled all possible connected subgraphs with three or five nodes. Notably, this covers the entire dataset and the complete diversity of conditions that the benchmarking kit offers.

• **Close:** We sampled all possible connected sub-graphs with edges that have a maximum geographic distance of 0.3 (euclidean distance of the nodes). By excluding long distances, the causal effect should be more pronounced. Notably, all subgraphs of this selection are included in "Random".

• **Random + 1:** We samples all possible connected sub-graphs that have two or four nodes. We then added another disconnected node to the graph. To prevent confounding, we sampled the random nodes from the coast and border area where we have several completely disconnected nodes.

• **Root cause:** We sampled all possible connected sub-graphs that have three or five nodes and where each has a maximum of one parent. With this, graphs are connected in the form of a single chain. We consider this useful for works on root-cause analysis (Ikram et al., 2022). Notably, all subgraphs of this selection are also included in "Random".

• **Confounder:** Probably, the most interesting set, we here selected sub-graphs with four or six nodes and where a single node has multiple children (while rare, these examples exist when rivers are naturally or artificially splitting). We then removed the node that has multiple children from the sample to simulate permanent hidden confounding scenarios.

• **Disjoint:** We sampled all possible connected sub-graphs that have five nodes and combine two of them into a single disjoint graph. To prevent connectivity, we chose to combine sampled with the largest possible geographical distance between them. With this, we aim to evaluate how causal discovery methods perform under a larger number of potential non-related variables.

The largest set, Random-5, holds more than 7500 subgraphs. The smallest set, Confounder-3, holds only 24 subgraphs. We report the results of this experiment in Table 3. Further, we report a full list of set sizes and alternative performance metrics in A.3. With some exceptions, we found that our naive baselines often achieve scores similar to actual causal discovery approaches. Concerning established causal discovery approaches, we found the linear Granger causal approach (VAR) to be the most reliant. For the "Root Cause" graph sets, we found that ordering the graphs according to their size (RP+N) can outperform all other causal discovery methods. Notably, while both CP and CDMI allow for non-linearity and, to some extent, seasonality, we found no evidence for their superiority over linear approaches on CausalRivers. Finally, while we found these graph characteristics to be a great start for comparison, many other characteristics could be explored (e.g., single-sink nodes, empty graphs, or causal pairs) to further unravel differences between causal discovery strategies.

Table 3: AUROC scores for Experiment Set 1. We mark the Top 2 performances in **green**. Null model refers to predicting no causal links, which achieves the smallest possible AUROC. †: CP networks are not able to process more than five variables. With some exceptions, Granger-based causal discovery (CD) approaches (VAR, Varlingam, and CDMI) achieve the most robust performance.

| | Structures: | | | | | | | | | | | |
|---|---|---|---|---|---|---|---|---|---|---|---|---|
| | | Close | | Root cause | | Random +1 | | Confounder | | Random | | Disjoint |
| | Method | 3 | 5 | 3 | 5 | 3 | 5 | 3 | 5 | 3 | 5 | 10 |
| *Naive Baselines* | RP | .80 | .76 | .76 | .70 | .74 | .73 | .62 | .66 | .79 | .75 | .75 |
| | RP+N | .72 | .62 | **.81** | .77 | .71 | .65 | .58 | .58 | .72 | .64 | .65 |
| | RP+B | .78 | .71 | .61 | .53 | .81 | .71 | .61 | .62 | .76 | .68 | .58 |
| | CC | .68 | .63 | .70 | .67 | .64 | .66 | .65 | .59 | .71 | .71 | .66 |
| | CC+C | .69 | .64 | .72 | .70 | .69 | .67 | .64 | .60 | .71 | .67 | .75 |
| | RPCC | .70 | .66 | .70 | .65 | .68 | .68 | .63 | .60 | .72 | .71 | .71 |
| | RPCC+N | .68 | .60 | .73 | .70 | .67 | .64 | .62 | .57 | .69 | .64 | .66 |
| | RPCC+B | .68 | .63 | .60 | .55 | .72 | .66 | .63 | .58 | .69 | .65 | .57 |
| | RPCC+C | .71 | .66 | .71 | .68 | .71 | .67 | .65 | .61 | .71 | .67 | .75 |
| | Null model | .50 | .50 | .50 | .50 | .50 | .50 | .50 | .50 | .50 | .50 | .50 |
| *CD Strategies* | VAR | **.81** | **.81** | .79 | .75 | .80 | **.79** | **.71** | **.72** | **.82** | **.80** | **.82** |
| | Varlingam | .79 | .77 | .77 | **.77** | **.84** | .79 | **.68** | .70 | .79 | .75 | **.83** |
| | Dynotears | .50 | .50 | .50 | .56 | .52 | .61 | .53 | .53 | .50 | .61 | .61 |
| | PCMCI | .64 | .62 | .70 | .74 | **.83** | .74 | .66 | .64 | .65 | .65 | .80 |
| | CDMI | **.81** | **.81** | .72 | .65 | .82 | **.80** | .63 | **.71** | **.80** | **.78** | .75 |
| | CP (Transf) | .60 | .65 | .62 | .68 | .80 | .72 | .56 | .56 | .60 | .65 | † |
| | CP (Gru) | .66 | .58 | .68 | .56 | .81 | .65 | .56 | .56 | .64 | .60 | † |

## 4.2 EXPERIMENT SET 2 - TIME-SERIES SUBSAMPLING

Given that the full time-series is very long (roughly 175k time steps for the original resolution), we were interested in whether selecting specific shorter, and hopefully informative, subsections might influence the performance of causal discovery algorithms. As a motivation, one might imagine that the complete time-series most likely holds sections with little innovation, displays annual patterns, and includes nonstationary windows with high amounts of change (such as RiversElbeFlood). To test whether providing only a subselection can improve in-the-wild causal discovery, we restricted the causal ground truth graph to the 42 nodes included in RiversElbeFlood. We then compared the causal discovery performance on the RiversElbeFlood graph with the performance on the full time-series and with the performance on a month with almost no recorded precipitation (Oktober 2021) in the selected region. Concerning subgraphs, we simply sampled all possible graphs with five nodes, equal to the sampling strategy "Random" from Experiment Set 1, and performed the same Hyperparameter optimization. We provide the results of this comparison in Figure 3a.

Interestingly, we found mixed results, as some methods seem to benefit from lower exogenous influences (PCMCI, "No rain") while others benefit from the additional strong distributional changes in "Flood". Especially noteworthy, Dynotears seems to struggle with the "No Rain" set while performing reasonably well on the other two sets. Our hypothesis here is that Dynotears, as a gradient-based method, struggles most with data that has little innovation. This could also be the reason why it showed the worst performance in Experiment Set 1, as there are more geographic locations with little elevation included. Next, we note that the performance on this subset of the ground truth causal graph was generally a little higher than in Experiment Set 1. We attribute this to the geographical location (more elevation) of the nodes included in RiversElbeFlood. We conclude that focusing on a proper time-series subselection strategy might be an interesting way forward to make causal discovery methods more robust in real-world applications, as it can strongly influence the performance of various methods. Furthermore, the property subselection seems to depend on the causal discovery method itself. Finally, we hope that these results encourage future works to put explicit effort into finding proper subselection strategies for the CausalRivers time-series that go beyond what we have shown here.

(a) Changes in method performance depending on the provided time-series data. We found mixed results with some pronounced differences, e.g., for Dynotears. Notably, both data regimes, *No rain* and *Flood*, only include four weeks of data while *Full TS* includes the complete five years.

| Method | *Full TS* | *No Rain* | *Flood* |
|---|---|---|---|
| RP | **0.78** | 0.78 | 0.70 |
| CC | **0.81** | 0.74 | 0.80 |
| RPCC | **0.81** | 0.79 | 0.74 |
| VAR | 0.85 | ↑**0.86** | 0.83 |
| Varlingam | 0.72 | 0.67 | ↑**0.74** |
| PCMCI | 0.60 | ↑**0.69** | 0.60 |
| Dynotears | 0.79 | 0.60 | ↑**0.80** |
| CDMI | **0.83** | † | † |
| CP | 0.65 | 0.66 | ↑**0.74** |

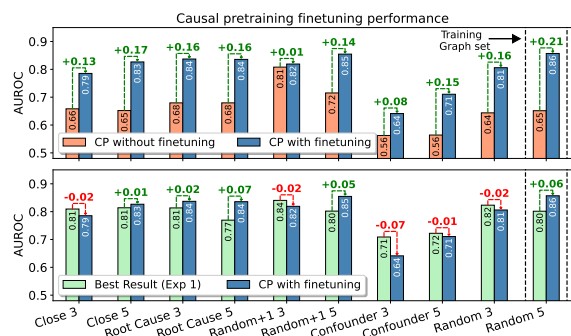

(b) Performance achieved through finetuning CP on domain samples. Finetuning CP networks with random five variable samples from *RiversBavaria* strongly boosts its performance, even on other more specified graph sets. This domain adaptation often leads to improvements over the best approaches from Experiment Set 1.

Figure 3: AUROC scores for Experiment Set 2 (a) and Experiment Set 3 (b). We mark increases and in performance with ↑. Further, the highest performance per method is marked in **bold**.

## 4.3 EXPERIMENT SET 3 - DOMAIN ADAPTION

As a final evaluation, we leveraged the fact that we include two distinct ground truth graphs to provide results on whether domain adaptation can be leveraged to improve causal discovery performance. As this area of research is not yet widely explored, we wanted to provide a first example of domain adaptation via Causal Pretraining (CP), a method that specifically allows for it, as causally pre-trained neural networks can be finetuned in a supervised manner. We, therefore, investigated whether the previously reported performance of CP on the RiversEastGermany dataset can be improved. To execute this, we leveraged RiversBavaria and sampled training and validation examples (identical to the strategy "Random-5") to finetune a pre-trained network provided by Stein et al. (2024). We performed a small Hyperparameter search, testing for different values of the learning rate, weight decay, batch size, time-series resolution, normalization, and the CP architecture. After training, we selected the network that achieved the highest F1 max on "Random 5" (a GRU on 12H resolution and no normalization) and evaluated it on all graph sets that were evaluated during Experiment Set 1. We report the results in Figure 3b and refer to [4] for details. We found that fine-tuning on random samples with 5 variables from RiversBavaria improves the performance of CP on all graph sets, suggesting that the learned adaptation is not restricted to the specific fine-tuning set. Further, this performance increase was sufficient to outperform the best causal strategy of Experiment Set 1 on 50% of the datasets. We take this as strong evidence that adapting casual discovery strategies to the target domain is highly beneficial and should be investigated thoroughly in future work.

## 5 CONCLUSION

In this paper, we presented CausalRivers, the largest in-the-wild causal discovery benchmarking kit for time series data to date. After motivating the need for such a benchmark by summarizing alternative datasets, we discussed the benchmarking kit and its unique challenges and opportunities. Further, we conducted a set of experiments, aiming at an evaluation of causal discovery approaches in real-world applications and an exploration of potential beneficial strategies. Our experiments showed, that many well-established causal discovery methods underperform in real-world applications and are even occasionally matched by simple but robust baseline strategies. With this, we conclude that more research is necessary, focusing on in-the-wild robustness, potentially through selecting relevant sections of a given time-series, and domain adaptation. We hope that this work provides the foundation for a benchmark-driven development of causal discovery methods and inspires the development of other benchmarking approaches.

---

[4]https://github.com/causalrivers/experiments

ACKNOWLEDGMENTS

We gratefully recognize the support of iDiv (German Centre of Integrative Biodiversity Research), which is funded by the German Research Foundation (DFG – FZT 118, 202548816). Gideon Stein is funded by the iDiv flexpool (No 06203674-22). Maha Shadaydeh is supported by the Deutsche Forschungsgemeinschaft (DFG, German Research Foundation) – Individual Research Grant SH 1682/1-1. Special thanks to the following German state and federal agencies for the provision of discharge data and their cooperation in this project:

- Thüringer Landesamt für Umwelt, Bergbau und Naturschutz
- Sächsisches Landesamt für Umwelt, Landwirtschaft und Geologie
- Landesamt für Umwelt Brandenburg
- Landesbetrieb für Hochwasserschutz und Wasserwirtschaft Sachsen-Anhalt
- Landesamt für Umwelt, Naturschutz und Geologie Mecklenburg- Vorpommern
- Wasserstraßen un Schifffahrtsverwaltung des Bundes
- Senatsverwaltung für Mobilität, Verkehr Klimaschutz und Umwelt
- Bayerisches Landesamt für Umwelt

We thank Tim Büchner for helping with online resources and Michel Besserve for providing valuable input on the first draft of the paper. Raw data sources fall under the Datenlizenz Deutschland.

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
