# A APPENDIX

## A.1 DATA ORIGINS, PREPROCESSING AND DATA STATISTICS

Table 4: Involved state and federal agencies that provided raw time-series discharge data and corresponding meta information. We here note the number of stations that remain in the final graph after preprocessing. "Freely available" denotes if the full five years of data are available from the corresponding web service.

| Agency Name | State | Abbreviation | Discharge stations | Freely available |
|---|---|---|---|---|
| Thüringer Landesamt für Umwelt, Bergbau und Naturschutz | Thuringia | T | 175 | ✗ |
| Sächsisches Landesamt für Umwelt, Landwirtschaft und Geologie | Saxony | S | 167 | ✓ |
| Landesamt für Umwelt Brandenburg | Brandenburg | BR | 145 | ✗ |
| Landesbetrieb für Hochwasserschutz und Wasserwirtschaft Sachsen-Anhalt | Saxony-Anhalt | SA | 92 | ✓ |
| Landesamt für Umwelt, Naturschutz und Geologie Mecklenburg-Vorpommern. | Mecklenburg–Western Pomerania | MV | 67 | ✗ |
| Wasserstraßen un Schifffahrtsverwaltung des Bundes | federal | BSCV | 12 | ✗ |
| Senatsverwaltung für Mobilität, Verkehr, Klimaschutz und Umwelt | Berlin | B | 8 | ✗ |
| Bayerisches Landesamt für Umwelt | Bavaria | BA | 494 | ✓ |

For our benchmarking kit, we fused several data sources that we aggregated from different state agencies in Germany and online resources. In Table 4, we list all agencies involved and some meta information. Concerning the causal ground truth graph, we relied on the Wikipedia pages of individual rivers[5], specifically in the German language, as these are very often more extensive. Other resources that were partly used are elevation services such as Meteo[6] and simply Google Maps [7] for manual quality control. Finally, we build on Hydrosheds[8] for visualizations. For further details, we refer to our repository[9].

As we, in many cases, receive raw time-series data from the state agencies without quality checks, we filter out stations that have more than 66% of missing data or that have no meta information available. Further, we remove doubled measurement stations and drop some stations that show clear signs of broken sensors (e.g., constant values for the majority of the time). Again, we provide the full preprocessing pipeline in our repository. Additional statistics can be observed in Figure 4, Figure 5, Figure 6, Figure 7 and Table 5. Notably, on average, time-series in "RiversEastGermany" include around 8% missing values, while for "RiversBavaria", only around 1% of values are missing.

---

[5]https://de.wikipedia.org/wiki/Elz_(Rhein)

[6]https://open-meteo.com

[7]https://www.google.com/maps/

[8]https://www.hydrosheds.org

[9]https://github.com/causalrivers

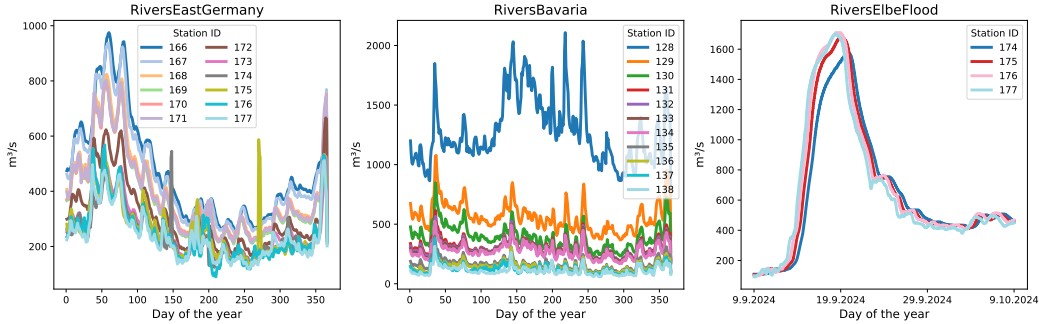

Figure 4: Left/Center: Annual discharge patterns (Mean over 5 years) of the biggest rivers (Elbe or Danube) in the three datasets. Notably, the Elbe shows a more pronounced annual cycle than the Danube, emphasizing distributional differences between the two datasets. Right: Discharge pattern of the Elbe river in the RiversElbeFlood dataset. A strong and sudden increase in discharge can be observed.

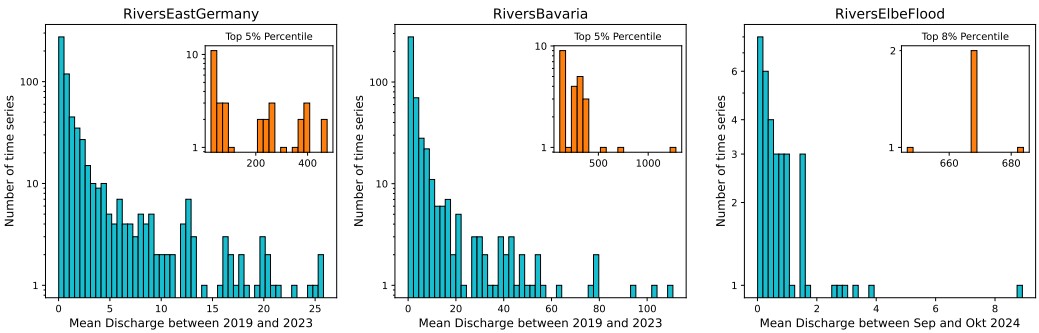

Figure 5: Distribution over the average discharge in the three CausalRivers time-series datasets. Notably, a few big rivers (e.g., Elbe, Danube, Oder) show vastly higher average discharges.

## A.2 HYPERPARAMETERS

Generally, we evaluate different time-series resolutions (15min,1H,6H,12H, 24H) and evaluate with normalized and unnormalized data. Additionally, we evaluate maximum lags (3 and 5) for VAR, PCMCI, Dynotears, and Varlingam. Further, for VAR, we evaluate whether considering absolute coefficient values is beneficial. For CP, we evaluated two architectures (a GRU and a Transformer). As CDMI does not provide default parameters, we rely on a number of Hyperparameters, that were selected based on the first 10 samples of the "random 5" graph set. We however only evaluate this single Hyperparameter combination on the full graph sets. Besides that, we rely on the default parameters of the specific implementations we use. All Hyperparameters, along with implementations and experiments, are documented in [10].

Table 5: Statistics of parent and child nodes in the ground truth causal graphs in CausalRivers. Most nodes have a single or no parent and a single child.

| Number of nodes | with $n$ predecessors | | | | | | | | | | | with $n$ successors | | | | |
|---|---|---|---|---|---|---|---|---|---|---|---|---|---|---|---|---|
| | 0 | 1 | 2 | 3 | 4 | 5 | 6 | 7 | 8 | 9 | 10 | 0 | 1 | 2 | 3 | 4 |
| RiversEastGermany | 296 | 206 | 108 | 26 | 16 | 5 | 6 | 1 | - | 1 | 1 | 44 | 596 | 24 | 1 | 1 |
| RiversBavaria | 257 | 110 | 64 | 34 | 15 | 2 | 7 | 2 | 3 | - | - | 19 | 462 | 11 | 2 | - |
| RiversFlood | 23 | 13 | 3 | - | 1 | - | - | - | - | 1 | 1 | 1 | 40 | 1 | - | - |

---

[10]https://github.com/causalrivers/experiments

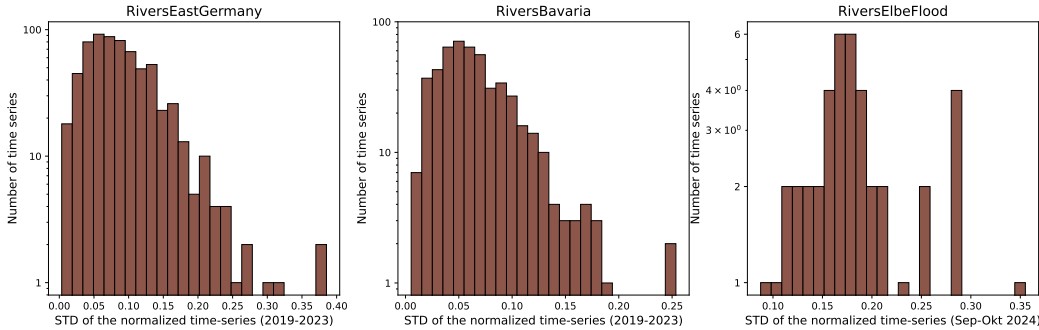

Figure 6: Distribution over the standard deviation of the normalized (0-1) time-series in the three CausalRivers time-series datasets. Depending on the geographical location and elevation changes, the amount of change in the time-series can vary.

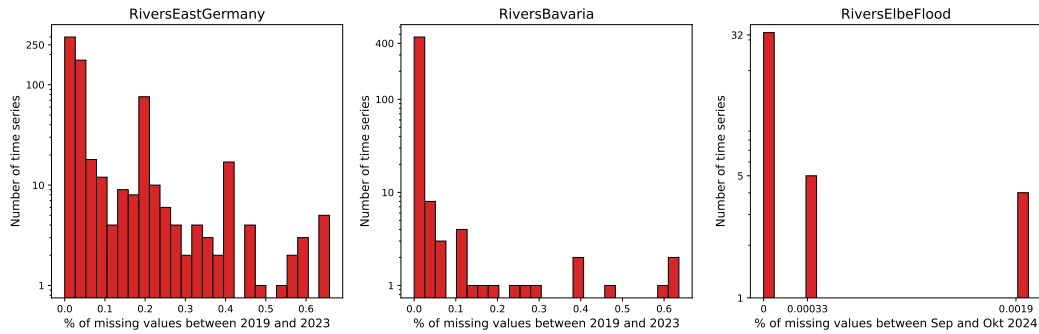

Figure 7: Distribution over the number of missing values per time-series in the three CausalRivers datasets (on a log scale). While RiversEastGermany includes more missing values, the total amount of missing values is below 10% in all cases.

## A.3    ADDITIONAL RESOURCES - EXPERIMENT SET 1 - 3

Table 6: A short list the number of samples that each of our evaluated graph sets holds.

| Graph set | Close | | Root cause | | Random +1 | | Random | | Confounder | | Disjoint |
|---|---|---|---|---|---|---|---|---|---|---|---|
| | 3 | 5 | 3 | 5 | 3 | 5 | 3 | 5 | 3 | 5 | 10 |
| Number of samples | 636 | 637 | 649 | 655 | 651 | 2790 | 1196 | 7521 | 24 | 361 | 7519 |

Here we include information on the amount of samples in each graph set that was used in Experiment Set 1 and 3 in Table 6. Further, we provide alternative threshold-free performance metrics for Experiment Set 1 in Table 7 and Table 8. Finally, we provide a depcition of the RiversFlood graph in Figure 8.

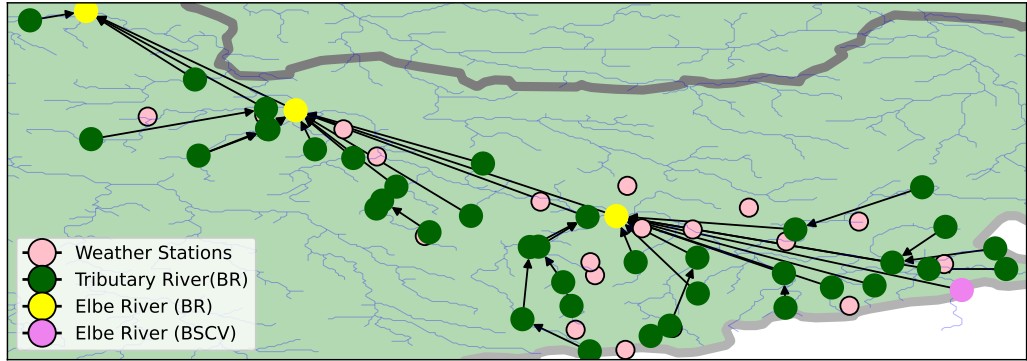

Figure 8: Causal ground truth graph of RiversFlood. This graph is a subset of RiversEastGermany. Weatherstations that were used to investigate the general precipitation levels are depicted in pink.

Table 7: F1 max scores for Experiment Set 1. We mark the Top 2 performances in green. The null model refers to predicting all causal links (as F1 is not defined without positive predictions), which achieves the smallest possible F1 max. †: CP networks are not able to process more than five variables. With some exceptions, Granger-based approaches (VAR, Varlingam, and CDMI) achieve the most robust performance.

| | Method | Close 3 | Close 5 | Root cause 3 | Root cause 5 | Random +1 3 | Random +1 5 | Confounder 3 | Confounder 5 | Random 3 | Random 5 | Disjoint 10 |
|---|---|---|---|---|---|---|---|---|---|---|---|---|
| *Naive Baselines* | RP | .75 | .53 | .71 | .48 | .48 | .42 | .66 | .48 | .74 | .52 | .29 |
| | RP+N | .68 | .46 | **.86** | **.67** | .55 | .43 | .55 | .45 | .69 | .49 | .36 |
| | RP+B | .73 | .57 | .50 | .33 | .61 | .50 | .56 | .47 | .71 | .53 | .24 |
| | CC | .65 | .44 | .66 | .46 | .44 | .39 | .65 | .43 | .67 | .49 | .25 |
| | CC+C | .67 | .49 | .70 | .57 | .52 | .46 | .60 | .45 | .68 | .52 | .53 |
| | RPCC | .66 | .47 | .64 | .45 | .50 | .42 | .62 | .44 | .67 | .51 | .30 |
| | RPCC+N | .65 | .44 | .73 | .58 | .52 | .43 | .60 | .44 | .67 | .49 | .38 |
| | RPCC+B | .65 | .48 | .55 | .36 | .56 | .44 | .60 | .43 | .65 | .49 | .25 |
| | RPCC+C | .68 | .52 | .68 | .54 | .55 | .47 | .62 | .47 | .68 | .52 | **.55** |
| | Null model | .50 | .34 | .50 | .33 | .29 | .26 | .54 | .36 | .50 | .33 | .16 |
| *CD Strategies* | VAR | .82 | **.68** | .80 | .62 | .77 | .60 | **.77** | .60 | **.83** | **.65** | .48 |
| | Varlingam | .81 | .65 | .79 | .64 | **.80** | .63 | .72 | **.61** | .81 | .63 | .48 |
| | Dynotears | .56 | .37 | .60 | .47 | .46 | .49 | .67 | .46 | .62 | .54 | .42 |
| | PCMCI | .68 | .48 | .71 | .57 | .77 | .55 | **.73** | .53 | .70 | .52 | .47 |
| | CDMI | **.82** | .67 | .73 | .50 | **.80** | **.63** | .72 | .59 | **.82** | .63 | .37 |
| | CP (Transf) | .68 | .53 | .69 | .55 | .77 | .55 | .65 | .49 | .68 | .53 | † |
| | CP (Gru) | .72 | .48 | .71 | .46 | .78 | .48 | .66 | .49 | .71 | .49 | † |

Table 8: Max Accuracy for Experiment Set 1. We mark the top 2 performances in **green**. The null model refers to predicting no causal links. †: CP networks are not able to process more than five variables. Simple Granger-based approaches (VAR, Varlingam) achieve the highest performance.

| | Method | Close | | Root cause | | Random +1 | | Confounder | | Random | | Disjoint |
|---|---|---|---|---|---|---|---|---|---|---|---|---|
| | | 3 | 5 | 3 | 5 | 3 | 5 | 3 | 5 | 3 | 5 | 10 |
| *Naive Baselines* | RP | .80 | .80 | .79 | .80 | .83 | .85 | .76 | .78 | .80 | .80 | .91 |
| | RP+N | .78 | .81 | **.90** | **.89** | .83 | .86 | .67 | .79 | .79 | .82 | .91 |
| | RP+B | .82 | .85 | .67 | .80 | .83 | .87 | .70 | .80 | .80 | .83 | .91 |
| | CC | .76 | .80 | .76 | .80 | .84 | .85 | .74 | .78 | .77 | .81 | .91 |
| | CC+C | .78 | .83 | .80 | .85 | .84 | .86 | .72 | .80 | .79 | .83 | .92 |
| | RPCC | .77 | .81 | .75 | .80 | .85 | .85 | .75 | .79 | .77 | .81 | .91 |
| | RPCC+N | .77 | .82 | .82 | .86 | .85 | .86 | .73 | .80 | .79 | .83 | .91 |
| | RPCC+B | .77 | .83 | .71 | .81 | .86 | .86 | .74 | .80 | .78 | .83 | .91 |
| | RPCC+C | .80 | .84 | .79 | .85 | .86 | .87 | .76 | .82 | .80 | .84 | .92 |
| | Null model | .67 | .80 | .67 | .80 | .83 | .85 | .63 | .78 | .67 | .80 | .91 |
| *CD Strategies* | VAR | **.87** | **.86** | .85 | .87 | .92 | .89 | .81 | .84 | **.87** | **.85** | **.93** |
| | Varlingam | .86 | .86 | .85 | .86 | **.93** | **.89** | **.81** | **.84** | .86 | .85 | .92 |
| | Dynotears | .71 | .81 | .75 | .83 | .87 | .88 | .74 | .81 | .75 | .85 | **.93** |
| | PCMCI | .71 | .80 | .78 | .83 | .92 | .87 | .80 | .81 | .77 | .83 | .91 |
| | CDMI | .83 | .84 | .78 | .81 | .91 | .87 | .78 | .81 | .81 | .82 | .91 |
| | CP (Transf) | .76 | .83 | .79 | .84 | .91 | .88 | .74 | .80 | .77 | .83 | † |
| | CP (Gru) | .79 | .81 | .80 | .81 | .92 | .87 | .76 | .80 | .79 | .82 | † |