# OpenReview forum: "CausalRivers - Scaling up benchmarking of causal discovery for real-world time-series"
_ICLR.cc/2025/Conference — ICLR 2025 Spotlight_

### Official Review · Reviewer_5VQx · 2024-10-22

**Soundness:** 3
**Presentation:** 3
**Contribution:** 4
**Rating:** 6
**Confidence:** 4

**Summary:**

This paper provides a large-scale benchmark dataset for causal discovery to estimate causal relationships between time series. The dataset consists of river flow measurements from East Germany and Bavaria, collected at 15-minute intervals from 2019 to 2023, and also includes flood data from the Elbe River area. Benchmark experiments using various causal discovery methods on the time series data revealed that a simple baseline method demonstrated the most consistent performance, outperforming many other causal discovery techniques. Additionally, the use of deep learning-based causal discovery methods showed a significant improvement in performance through domain adaptation across different datasets. This dataset represents an unprecedented benchmark for causal discovery in time series data and is expected to greatly contribute to advancing research in this field.

**Strengths:**

This paper presents a large-scale benchmark dataset for causal discovery to estimate causal relationships between multivariate time series. The dataset provides an unprecedented scale of ground-truth data for causal discovery in time series, and it is expected to contribute to the advancement of research in this area significantly. Furthermore, the benchmark experiments conducted using this dataset revealed that a simple baseline method outperformed many other causal discovery techniques, offering important insights into the field, which has traditionally relied on evaluations using artificial data or simpler real-world datasets.

**Weaknesses:**

The benchmark dataset provided in this paper is of unprecedented scale and will undoubtedly contribute significantly to the field of causal discovery. However, the task may be somewhat simplified because the dataset is tied to geographical information. Predicting causal relationships between nearby river basins is relatively easy, meaning that causal discovery might only need to focus on a limited subset of the series. This could slightly diminish the dataset's overall value. The experiments seem to focus only on causal discovery within subsets of the time series. A broader evaluation of causal discovery across all series could lead to a more in-depth discussion.

Additionally, there are a few minor typographical errors. For instance, in Section 3.2.1, the phrase "benchmarking kid" should be corrected to "benchmarking kit." Moreover, variables with upper bars are used in the same section without explanation, and the notation should be clearly defined.

**Questions:**

As noted in Table 1, there is still a lack of sufficient benchmark datasets for causal discovery in non-time-series domains, even for real-world data. Would it be feasible to construct similar benchmark datasets for causal discovery in non-time-series settings?

---

> ### Author Response · Authors · 2024-11-20
>
> First of all, thank you for this review and the valuable input.  In the following, we address the specific issues you mentioned in the order they were raised and note where we improved the paper accordingly.
>
> **Concerning the potential simplification through geographical locations**
>
> Geographically close stations do not necessarily imply more straightforward causal discovery.  First, geographical information is provided but not intended to be used in the causal discovery. Second, The causal effect strength and the elevation differences also play a prominent role. For example, it is very hard to determine the causal effect of a river carrying little water on a large body of water, such as the Elbe River, even if they are geographically close. We revised  Section 3.3 (geographical realities) to address your point. Notably, the experiments on graph sets “close 3” and “close 5” (experiment section 1) seem to confirm this intuition as methods typically perform a little (but not much) better on these subgraphs than in comparison to “random 3” and “random 5”.
>
> **Concerning the issue that a focus on a limited subset of a series might be enough for causal discovery**
>
> A specific time series subsection could be used to perform accurate causal discovery. However, selecting the suitable section is non-trivial, as shown in the results of Experiment Set 2. Here, we evaluate some potential candidate subsections but observe no clear-cut improvements. Nonetheless, this is an important point as an appropriate subselection is a way to improve causal discovery algorithms' performance in real-world applications. We are aware of some more advanced strategies that could be used to perform time-series subselection for causal discovery purposes, such as [1] or [2]. However, such strategies have not been a central topic of discussion in the causal discovery literature so far. We added this in the paper's final section as a potential work using our benchmark kit. We also adapted the motivation of Experiment Set 2 to emphasize this point a little more.
>
> **Concerning the focus on a subset of time-series**
>
> We should have noted that more explicitly. The subgraph sets “random 3” and “random 5” include all possible subgraphs with 3 or 5 directly connected nodes, effectively evaluating the complete graph and all available time series. Here, we want to emphasize that this kind of analysis covers the full diversity of the dataset's geographical conditions. Next to this, other sets of subgraphs are used to dissect method performance over different graphical structures. We revised the paper by updating the description (4.1.1) and Table 2.
> Further, it would be interesting also to test causal discovery algorithms on larger groups of variables (12+). While this is possible by using the code of our benchmarking kit’s repository, causal discovery algorithms often scale poorly with the number of variables. Thus, we refrain from presenting results on subgraphs with more than 12 nodes.
>
> We hope that we have covered the weaknesses you raised sufficiently. If some uncertainties remain, we are happy to discuss them further.
> Finally, we reviewed the text to correct spelling mistakes and improve naming consistency. All the points that you mentioned were corrected. Thank you for pointing them out.
>
> **Concerning your question about the non-time-series settings**
>
> Yes. Other similar benchmarking datasets, especially on non-time-series data,  should be feasible. We would be happy to see such work in the future and note this accordingly in the final section.
>
> [1] Ahmad, Wasim, et al. "Deep‐learning based causal inference: A feasibility study based on three years of tectonic‐climate data from Moxa geodynamic observatory." Earth and Space Science 11.10 (2024): e2023EA003430.
>
> [2] Deldari, Shohreh, et al. "Time series change point detection with self-supervised contrastive predictive coding." Proceedings of the Web Conference 2021. 2021.

---

### Official Review · Reviewer_TrPo · 2024-11-02

**Soundness:** 3
**Presentation:** 3
**Contribution:** 3
**Rating:** 8
**Confidence:** 4

**Summary:**

This paper introduces the CausalRivers benchmarking kit for comprehensive time series data analysis of river discharge in Germany. With high-frequency sampling every 15 minutes, the dataset includes data from a significant flood event on the Elbe River, making it suitable for testing causal discovery under real-world conditions with distributional shifts. The kit supports a range of causal discovery methods, from traditional approaches like Granger causality to advanced models like PCMCI, VARLINGAM, Dynotears, and CDMI, and allows sampling of subgraphs for diverse benchmarking cases.
The experimental evaluation uses data from 666 and 494 stations across eastern Germany and Bavaria, respectively, from 2019 to 2023.
The results revealed that while advanced methods struggle with real-world complexities like non-stationarity and high dimensionality, simple baselines performed robustly in identifying causal links. Domain adaptation via fine-tuning also showed performance gains.

**Strengths:**

The paper’s main strength is the extensive, realistic testing of causal discovery methods in the wild, which provides valuable insights into performance in a complex problem. CausalRivers supports various graph structures and sampling techniques, making it adaptable for different scenarios. It is obvious that benchmarks are important, and CausalRivers’ advanced methods allow for comprehensive performance comparisons across various scenarios.

**Weaknesses:**

Although this paper has an intensive assessment of models, its main weakness is that it is a technical benchmark. That is, the dataset’s complexity, along with various methods, may present implementation challenges for causal discovery researchers, and thus, it offers scientific opportunities. However, the scientific insight from the paper is missing (unless ICLR has changed its structure and now accepts also non-research but technical contributions, in which case this paper would be a good fit)

**Questions:**

NA

---

> ### Author Response · Authors · 2024-11-19
>
> Thank you for the time you put into assessing our paper and also for noting that extensive and realistic benchmarks are important.
>
> Concerning your raised issue about the ICLR main track structure, ICLR 2025 accepts submissions in the area of “datasets and benchmarks,” which we feel is also the proper category for this paper. To clear up this confusion, we checked the historical development of this and, as you say, it seems that last year was the first year where such submissions were accepted. Noteworthy, we were, in fact, inspired by last year's invited talk, “The emerging science of benchmarks“ (https://iclr.cc/virtual/2024/invited-talk/21799), to construct this benchmark as we felt this to be missing in the space of causal discovery. While benchmarks are, of course, always a little backloaded concerning scientific insights, they are the foundation for such and, with this, to the best of our knowledge, are also deemed important by ICLR.
>
> Further, we not only present the benchmark kit itself but also provide some experiments that have implications for causal discovery in-the-wild. Experiment Sets 2 and 3 (on domain adaptation and time-series subselection) are concerned with topics that have been sparsely discussed so far (as such procedures are only properly testable with a large amount of data available). We consider them to be scientific insights. To make this more transparent, we updated the motivations for both experiment sets.

---

> > ### Comment · Reviewer_TrPo · 2024-12-03
> >
> > Thank you for your answer. As I previously stated, this paper offers scientific opportunities given the extensive evaluations and the data presented. I independently verified the adequacy of the benchmarks for ICLR and thus will increase my rating accordingly.

---

### Official Review · Reviewer_qr8D · 2024-11-04

**Soundness:** 4
**Presentation:** 4
**Contribution:** 4
**Rating:** 8
**Confidence:** 4

**Summary:**

The authors present a high resolution dataset of river discharges that covers two large geographical regions in Germany with the purpose of benchmarking causal discovery algorithms. The presented dataset is interesting and presents a significant improvement over existing benchmarks in terms of scale and resolution and has the potential to be a significantly contribution to the development of causal discovery algorithms.

**Strengths:**

- The dataset is interesting and improves over existing benchmarks both in terms of size and resolution.
-  The authors provide a reliable baseline for the causal relations in the dataset.
-  The paper is well written and easy to follow.
-  The authors provide an extensive set of experimental baselines along with multiple software tools for analysing and processing the dataset.
- The authors commit to publishing the full pipeline used to construct the data set ensuring reproducibility.

**Weaknesses:**

- The authors could have discussed the completeness and reliability of the dataset, such as data quality checks and missing data handling more extensively.
- Allowing the reviewers access the dataset and software resources would have been beneficial but is understandable given the nature of the article.
- Although the authors mention that the data is compiled the from multiple sources these are not specified in the paper, providing a list of specific data sources would enhance transparency and reproducibility.

**Questions:**

- see weaknesses above

---

> ### Author Response · Authors · 2024-11-19
>
> Dear reviewer, thank you for your review and especially the raised weaknesses.
> We comment on these in the following, as you noted them as potential questions:
>
> - Concerning completeness and reliability:
> Thank you for noting this. We agree that the paper would benefit from discussing these details. We, therefore, added a section in the appendix that covers topics such as preprocessing, data quality, and missing data.
>
> - Data & software resources
> will, of course, be uploaded as soon as the paper is accepted and in time for the conference. Nonetheless, we uploaded a stripped-down version, excluding data as an anonymous repository to provide resources if needed here:
> https://anonymous.4open.science/r/CR_strip_down-B626
>
> - Concerning Data sources:
> Thank you for pointing this out. We added additional descriptions to the colors in Fig. 1 to note their origin (the state agency that provided them). Further, we added a section in the appendix that describes how the data sources were accessed. Further, we will mention all the state agencies in the acknowledgment section of the paper (which was omitted due to blind review).

---

> > ### Comment · Reviewer_qr8D · 2024-11-27
> >
> > I would like to thank the authors for their reply and the modifications they made to the manuscript which adequately address the concerns I had initially raised.

---

### Author Response · Authors · 2024-11-20

Dear Reviewers,

We are grateful for the time and effort you took to review our submission and appreciate the constructive feedback that you provided. We are pleased to hear that the reviewers appreciate the effort that we put into constructing this benchmarking kit. Further, we are happy to note that it was jointly recognized that our work could significantly impact the field of causal discovery.

We would like to inform you that we pushed a stripped-down version of our benchmarking kit here:
https://anonymous.4open.science/r/CR_strip_down-B626

We hope that this will help with any open issues that remain and look forward to answering any remaining questions or concerns.

---

### Meta-Review · Area_Chair_QuAv · 2024-12-20

**Metareview:**

This paper proposes a high-resolution dataset of river discharges covering two large geographical regions in Germany, designed to benchmark causal discovery algorithms. The idea is considered innovative and impactful but the paper has some weaknesses, such as limited discussion on data completeness and reliability. Fortunately, the authors have addressed the main issues in the responses.

**Additional Comments On Reviewer Discussion:**

The idea is considered innovative and impactful (qr8D), extensive and realistic (TrPo), and unprecedented in scale (5VQx). However, the paper has some weaknesses, such as limited discussion on data completeness and reliability (qr8D), a lack of scientific insight (TrPo), and simplified task complexity due to geographical constraints (5VQx). Fortunately, the authors have addressed the main issues in the responses.

---

### Decision · Program_Chairs · 2025-01-22

Accept (Spotlight)